# Social isolation and psychological distress among southern U.S. college students in the era of COVID-19

**Danielle Giovenco**[1]*, **Bonnie E. Shook-Sa**[2], **Bryant Hutson**[3], **Laurie Buchanan**[3], **Edwin B. Fisher**[4], **Audrey Pettifor**[1]

**1** Department of Epidemiology, Gillings School of Global Public Health, University of North Carolina at Chapel Hill, Chapel Hill, NC, United States of America, **2** Department of Biostatistics, Gillings School of Global Public Health, University of North Carolina at Chapel Hill, Chapel Hill, NC, United States of America, **3** Institutional Research and Assessment, University of North Carolina at Chapel Hill, Chapel Hill, NC, United States of America, **4** Peers for Progress and Department of Health Behavior, Gillings School of Global Public Health, University of North Carolina at Chapel Hill, Chapel Hill, NC, United States of America

* danielle.giovenco@emory.edu

**Data Availability Statement:** Data may contain identifying or sensitive participant information. To preserve participant confidentiality, these data cannot be shared publicly. The Principal

## Abstract

### Background

College students are at heightened risk for negative psychological outcomes due to COVID-19. We examined the prevalence of psychological distress and its association with social isolation among public university students in the southern United States.

### Methods

A cross-sectional survey was emailed to all University of North Carolina-Chapel Hill students in June 2020 and was open for two weeks. Students self-reported if they were self-isolating none, some, most, or all of the time. Validated screening instruments were used to assess clinically significant symptoms of depression, loneliness, and increased perceived stress. The data was weighted to the complete student population.

### Results

7,012 completed surveys were included. Almost two-thirds (64%) of the students reported clinically significant depressive symptoms and 65% were categorized as lonely. An estimated 64% of students reported self-isolating most or all of the time. Compared to those self-isolating none of the time, students self-isolating some of the time were 1.78 (95% CI 1.37, 2.30) times as likely to report clinically significant depressive symptoms, and students self-isolating most or all of the time were 2.12 (95% CI 1.64, 2.74) and 2.27 (95% CI 1.75, 2.94) times as likely to report clinically significant depressive symptoms, respectively. Similar associations between self-isolation and loneliness and perceived stress were observed.

### Conclusions

The prevalence of adverse mental health indicators among this sample of university students in June 2020 was exceptionally high. University responses to the COVID-19

Investigator of this study, Audrey Pettifor (apettif@email.unc.edu), or the University of North Carolina-Chapel Hill's Office of Institutional Research and Assessment (oira@unc.edu) may be contacted with requests to access these data.

**Funding:** This research was supported by the University of North Carolina at Chapel Hill and grant F31MH119965 (PI: Giovenco) of the National Institutes of Mental Health. The funders had no role in study design, data collection and analysis, decision to publish, or preparation of the manuscript. There was no additional external funding received for this study.

**Competing interests:** The authors have declared that no competing interests exist.

pandemic should prioritize student mental health and prepare a range of support services to mitigate mental health consequences as the pandemic continues to evolve.

## Introduction

The COVID-19 pandemic has resulted in poor mental health outcomes among diverse populations globally [1]. In June 2020, a survey assessing mental health challenges related to COVID-19 among a nationally-representative sample of adults in the United States (U.S.) found that young adults (18–24 years) reported the highest prevalence of symptoms of depression (52%) and anxiety (49%) compared to any other age group [2]. The prevalence of adverse mental health outcomes among those aged 25–44 years was also high, with the prevalence of adverse mental health outcomes decreasing as age increased.

College and university students are a unique group of young adults facing academic, interpersonal, and environmental stressors who have historically experienced high rates of mental health distress compared to the general population [3–5]. A systematic review conducted in 2013 estimated that the mean weighted prevalence of depression among college students was 30.6% [4]. Research has shown that the prevalence of mental health conditions among college students is increasing, and college students are also more likely to seek mental health services, likely contributing to these findings [6, 7]. However, the majority of college students who struggle with mental health conditions still do not seek care [8], meaning many students are not diagnosed and do not receive needed treatment.

There is a growing body of literature describing college student mental health in the context of the COVID-19 pandemic [9–15]. Many of these studies from the U.S. have shown a high prevalence of depression, stress, and anxiety that are particularly pronounced among women, low-income students, and minority students [11–15]. Multiple stressors have been theorized as contributors to the increased levels of depression, stress, and anxiety that have been observed, including worry about one's own health and the health of loved ones, difficulty in concentrating, increased concerns about academic performance, disruptions to sleeping patterns, and decreased social interactions due to physical distancing [15].

During periods of social isolation, individuals are prone to experiencing heightened levels of psychological distress [1]. Despite this knowledge, there is a lack of research quantitatively examining the association between social isolation and psychological distress outcomes among college students in the context of COVID-19. Further, research on the impact of the COVID-19 pandemic on college student mental health is often limited to small samples that are not representative of larger student populations. In the present study, we aimed to 1) characterize the prevalence of symptoms of psychological distress among a large, weighted sample of public university students in the southern U.S. and 2) examine the link between social isolation and several psychological distress outcomes.

## Methods

### Study overview

The University of North Carolina at Chapel Hill (UNC-CH) is a large public research university with 21,223 undergraduate and 12,016 graduate and professional degree-seeking students enrolled at the time of this study. In an initial response to COVID-19, UNC-CH significantly reduced operations on March 20, 2020, requiring students to vacate campus housing by March 21st, and shifted to remote instruction on March 23rd.

A cross-sectional survey aimed at assessing student knowledge, attitudes, and behaviors related to SARS-CoV-2 was emailed to all UNC-CH undergraduate, graduate, and professional students on June 8th, 2020. The survey was open for two weeks until June 23rd. Students had to be at least 18 years of age to be eligible to participate. Prior to starting the survey, students interested in participating followed a link to read and sign an informed consent form. The survey consisted of 47 questions, many of which had several parts, and incorporated multiple-choice, multiple-answer, and open-ended response questions. Questions related to COVID-19 were drawn from similar surveys or were based on our own design. We also included several validated measures to assess student well-being. The survey took approximately 30 minutes to complete. Students who completed at least 75% of the survey were entered into a drawing for one of fifty $50 gift certificates.

Demographic and student data (e.g., graduate or undergraduate student type, full or part-time student status, residency, and U.S. citizenship) was provided by the university registrar and linked to survey responses prior to data de-identification for analyses. The Institutional Review Board (IRB) of the UNC-CH Office of Human Research Ethics approved the study procedures. Electronic consent was obtained from all participants.

## Measures

**Social Isolation exposures.** For our primary exposure, participants were asked, "to what extent are you self-isolating?". Answer options included: 1) all of the time–I am staying at home nearly all the time; 2) most of the time–I only leave my home to buy food and other essentials; 3) some of the time–I have reduced the amount of times I am in public spaces, social gatherings, or at work; 4) and none of the time–I am doing everything I normally do.

Several additional questions were used to assess social isolation. In a measure aimed at assessing attitudes towards COVID-19 prevention and control measures, participants were asked, "How much you disagree or agree with the following statements: 1) I avoid crowded areas and 2) I avoid getting together with people who are not part of my household". In a measure aimed at assessing behavioral changes related to COVID-19, participants were asked, "To what extent do you agree with each of the following statements about your behavior in the past month as a result of the new coronavirus: 1) I stayed at home and 2) I did not attend social gatherings". For both measures, participants were asked to rate their behavior on a 5-point Likert scale ranging from strongly disagree to strongly agree.

**Psychological distress outcomes.** Scores for each psychological distress outcome variable were calculated only for participants with complete data for all measure items. When available, we dichotomized outcomes using clinically significant cutpoints to improve the interpretability of our findings.

The 10-item Center for Epidemiological Studies Depression Scale (CES-D-10) is a widely used questionnaire assessing clinically significant depressive symptoms in the previous week [16]. It includes three items on depressed affect, five items on somatic symptoms, and two on positive affect. Likert scale options for each item range from "rarely or none of the time" (score of 0) to "all of the time" (score of 3). Scoring is reversed for items based on statements of positive affect. The total score is the sum of 10 items (possible range = 0–30). Based on previous studies [16], a total score equal to or above 10 was used to identify individuals reporting clinically significant symptoms of depression.

The 3-item Loneliness Scale (UCLA-3) is a questionnaire developed from the Revised UCLA Loneliness Scale assessing feelings of loneliness or social isolation in the previous month [17]. Each question was rated on a 3-point scale: 1 = hardly ever; 2 = some of the time; 3 = often. All items are summed to give a total score, with higher scores indicating greater

degrees of loneliness (possible range = 3–9). Consistent with previous research, we categorized individuals with total scores equal to or above 6 as lonely [18, 19].

The 4-item Perceived Stress Scale (PSS-4) is a questionnaire that assesses the degree to which situations in one's life over the previous month are appraised as stressful [20, 21]. Each question was rated on a 5-point scale: 0 = never, 1 = almost never, 2 = sometimes, 3 = fairly often, 4 = very often. Scores are obtained by reverse coding two positive items and then summing scores across all four items, with higher scores indicating a higher perceived stress level (possible range = 0–16). For analysis, total scores were dichotomized at the unweighted sample median, with total scores at or below the median indicating lower perceived stress and scores above the median indicating greater perceived stress.

### Analysis

Our analysis sample included only students who completed the survey, regardless of whether items were skipped. To examine the potential for bias due to excluding persons who started but did not complete the surveys, we compared the distribution of demographic characteristics, self-isolation, and psychological distress outcomes for survey completers and all survey respondents. Then, to adjust for student nonresponse (or partial response), we used iterative proportional fitting (i.e., raking) methods to weight the sample of survey completers to the marginal distributions of the UNC-CH student population by age category (<21, 21–24, 25–34, and ≥35 years), race and ethnicity (White, Black or African American, Asian, Hispanic of any race, and other or multiple races), gender, and student type (undergraduate and graduate or professional). University registrar data for all eligible students enrolled in June 2020 were used to create marginal control totals that were entered into the raking algorithm [22]. Iterative weight adjustments continued until the weighted margins differed from population margins by <1% for each raking variable.

We described the unweighted and weighted sample distributions for demographic and student characteristics provided by the UNC-CH registrar. All results are presented weighted, with their unweighted counterparts included in the S1 File. First, the proportion of students who self-reported that they were self-isolating most or all of the time (vs. some or none of the time) was described for each level of demographic and student characteristics, and Wald chi-square tests were used to compare the proportion of students who reported self-isolating across levels of the covariates. Then, we described the overall prevalence of social isolation variables, clinically significant depressive symptoms, and loneliness, as well as the distribution of perceived stress. The proportion of students with each psychological distress outcome and greater perceived stress were compared across age, race/ethnicity, gender, and student type categories. We assessed the internal reliability of each outcome measure (CES-D-10, UCLA-3, PSS-4) using Cronbach's alpha.

Log-binomial regression was used to calculate prevalence ratio (PR) estimates for associations between social isolation and psychological distress. Robust variance estimators were used for weighted regression models. For our primary exposure, we estimated the relative prevalence of each psychological distress outcome among participants who reported self-isolating some, most, or all of the time versus none of the time (referent), and a Cochran-Armitage test for trend was conducted ($\alpha$ = .05). For each additional social isolation exposure, psychological distress prevalence among participants who selected "somewhat agree" or "strongly agree" was compared to participants who selected "somewhat disagree" or "strongly disagree" (referent). Statistical analyses were conducted in SAS 9.4 (Cary, NC).

### Results

A total of 33,239 UNC-CH students were emailed the survey. Among these, 9,531 students started the survey (29% response), of whom 7,012 (74%) completed the survey and were

**Table 1. Demographic and student characteristics.**

| Characteristic | Unweighted sample | Weighted sample | Self-isolating most or all of the time (versus some or none of the time) | | |
|---|---|---|---|---|---|
| | N (%) | N (%) | Weighted N (%) | 95% CI | p-value[c] |
| **Total N** | 7012 | 33239 | 21251 (64%) | 62.8%, 65.2% | |
| **Age** | | | | | |
| <21 years | 3656 (52%) | 15497 (47%) | 9048 (58%) | 56.8%, 60.1% | |
| 21–24 years | 1712 (24%) | 8548 (26%) | 5437 (64%) | 61.4%, 66.1% | |
| 25–34 years | 1338 (19%) | 7066 (21%) | 5171 (73%) | 70.7%, 75.7% | |
| ≥35 years | 305 (4%) | 2127 (6%) | 1595 (75%) | 70.0%, 80.3% | < .001 |
| **Race/Ethnicity** | | | | | |
| White | 4422 (66%) | 19304 (61%) | 11078 (57%) | 56.0%, 59.0% | |
| Black or African American | 406 (6%) | 2639 (8%) | 1921 (73%) | 68.1%, 77.4% | |
| Asian | 1016 (15%) | 5323 (17%) | 4215 (79%) | 76.6%, 81.8% | |
| Hispanic of any race | 541 (8%) | 2885 (9%) | 1972 (68%) | 64.4%, 72.5% | |
| Other[a] or multiple races | 365 (5%) | 1724 (5%) | 1117 (65%) | 59.7%, 69.9% | < .001 |
| **Gender** | | | | | |
| Female | 4999 (71%) | 19425 (58%) | 12602 (65%) | 63.6%, 66.3% | |
| Male | 2007 (29%) | 13789 (42%) | 8628 (63%) | 60.5%, 64.8% | .080 |
| **Student type** | | | | | |
| Undergraduate student[b] | 4754 (68%) | 21223 (64%) | 12780 (60%) | 58.8%, 61.8% | |
| Graduate/prof student | 2258 (32%) | 12016 (36%) | 8471 (71%) | 68.6%, 72.5% | < .001 |
| **Full-time status** | | | | | |
| Part-time | 2206 (31%) | 10359 (31%) | 6325 (61%) | 58.9%, 63.2% | |
| Full-time | 4805 (69%) | 22879 (69%) | 14925 (65%) | 63.9%, 66.8% | .001 |
| **Residency** | | | | | |
| In-state | 5237 (75%) | 24176 (73%) | 15187 (63%) | 61.6%, 64.3% | |
| Out-of-state | 1768 (25%) | 9028 (27%) | 6044 (67%) | 64.6%, 69.3% | .003 |
| **Citizenship** | | | | | |
| U.S. citizen | 6376 (91%) | 29714 (90%) | 18448 (62%) | 60.9%, 63.4% | |
| Non-U.S. citizen | 628 (9%) | 3485 (10%) | 2771 (80%) | 76.2%, 82.9% | < .001 |

Estimates exclude 1 (.01%) participant missing age, 262 (3.7%) missing race/ethnicity, 6 (.09%) missing gender, 1 (.01%) missing full-time status, 7 (.10%) missing residency, and 8 (.11%) missing citizenship.

[a]Includes 'American Indian or Alaskan Native' or 'Native Hawaiian or other Pacific Islander'.

[b]Includes 18 post-baccalaureate students.

[c]Wald chi-square test comparing the percent self-isolating most or all of the time across levels of covariates.

included in the analysis sample. The median age of students who completed the survey was 20 years (interquartile range (IQR) = 19–24). A comparison of survey completers (n = 7,012) and all survey respondents (n = 9,531) on demographic characteristics and primary exposure and outcome variables revealed no substantive differences between groups (Table 1 in S1 File). The distribution of survey completers was also largely similar to the distribution of UNC-CH students for the demographic and student characteristic domains examined, with the exception of gender (Table 1). Thus, differences between weighted and unweighted estimates were minimal. The weighted student population was predominantly <25 years of age (73%), female (58%), non-Hispanic White (61%), and enrolled in full-time study (69%). Sixty-four percent self-reported they were self-isolating most or all of the time.

Self-isolation varied by demographic and student characteristics. Students 25–34 (73%) and ≥35 (75%) years were more likely to report they were self-isolating most or all of the time than

**Table 2. Weighted distribution of social isolation exposure variables (N = 33,239).**

| Social isolation variables | N (%) | 95% CI |
|---|---|---|
| **Self-isolation[a]** | | |
| None of the time | 659 (2%) | 1.6%, 2.3% |
| Some of the time | 11288 (34%) | 32.8%, 35.2% |
| Most of the time | 16927 (51%) | 49,8%, 52.2% |
| All of the time | 4324 (13%) | 12.2%, 13.9% |
| **I avoid crowded areas[b]** | | |
| Somewhat/strongly disagree | 1065 (3%) | 2.9%, 3.8% |
| Somewhat/strongly agree | 30890 (97%) | 96.2%, 97.1% |
| **I avoid getting together with people who are not part of my household[b]** | | |
| Somewhat/strongly disagree | 6416 (21%) | 20.4%, 22.5% |
| Somewhat/strongly agree | 23459 (79%) | 77.5%, 79.6% |
| **I stayed home[c]** | | |
| Somewhat/strongly disagree | 2352 (7%) | 6.6%, 7.9% |
| Somewhat/strongly agree | 29931 (93%) | 92.1%, 93.4% |
| **I did not attend social gatherings[c]** | | |
| Somewhat/strongly disagree | 3046 (10%) | 8.8%, 10.3% |
| Somewhat/strongly agree | 28724 (90%) | 89.7%, 91.2% |

Estimates exclude 8 participants missing self-isolation (.12% of weighted observations); 21 missing and 245 who responded "neither agree nor disagree" for "I avoid crowded areas" (3.9% of weighted observations); 16 missing and 705 who responded "neither agree nor disagree" for "I avoid getting together with people who are not part of my household" (10.1% of weighted observations); 14 missing and 187 who responded "neither agree nor disagree" for "I stayed home" (2.9% of weighted observations); and 25 missing and 287 who responded "neither agree nor disagree" for "I did not attend social gatherings" (4.4% of weighted observations).

[a]To what extent are you self-isolating?

[b]Please indicate how much you disagree or agree with the following statements.

[c]To what extent do you agree with each of the following statements about your behavior in the past month as a result of the new coronavirus?

those <21 (58%) and 21–24 (64%) years. Further, Asian and Black/African American students were most likely to be self-isolating (79% and 73%, respectively) than any other race group, and White race students were least likely (57%). Those who reported self-isolating were also more likely to be graduate/professional students, full-time students, out-of-state residents, and non-U.S. citizens (Table 1). The majority of students agreed or strongly agreed with statements that they were avoiding crowded areas (97%) and not getting together with people outside of their households (79%), and in the previous month, they stayed home (93%) and did not attend social gatherings (90%) (Table 2).

Almost two-thirds (64%) of the UNC-CH students reported clinically significant depressive symptoms on the CES-D-10, and 65% were categorized as lonely on the UCLA-3 (Table 3). Further, 41% of students reported levels of perceived stress on the PSS-4 above the unweighted sample median score of 8, indicating greater perceived stress. For the weighted sample, the median CES-D-10 score was 12 (IQR = 7–17), the median UCLA-3 score was 6 (IQR = 5–8), and the median PSS-4 score was 8 (IQR = 6–10). All three psychological distress scales had good or acceptable internal consistency (CES-D-10 $\alpha$ = .87, UCLA-3 $\alpha$ = .78, and PSS-4 $\alpha$ = .76). Missing data for psychological distress variables in the analytic sample were minimal, with <2% of students missing data for measure items.

Psychological distress outcome prevalence varied by gender, race, age, and student type. Women were more likely than men to report clinically significant depressive symptoms (71%

**Table 3. Weighted distribution of psychological distress outcome variables (N = 33,239).**

| Mental health variables | N (%) | 95% CI |
|---|---|---|
| **CES-D-10** | | |
| Non-clinically significant depressive symptoms (score <10) | 11920 (36%) | 35.3%, 37.7% |
| Clinically significant depressive symptoms (score ≥10) | 20759 (64%) | 62.3%, 64.7% |
| **UCLA-3** | | |
| Not lonely (score = 3–5) | 11702 (35%) | 34.1%, 36.5% |
| Lonely (score = 6–9) | 21403 (65%) | 63.5%, 65.9% |
| **PSS-4**[a] | | |
| Lower stress (score = 0–8) | 19389 (59%) | 57.5%, 60.0% |
| Greater stress (score = 9–16) | 13610 (41%) | 40.0%, 42.5% |

Estimates exclude 114 participants missing a CES-D-10 score (1.7% of weighted observations), 24 missing a UCLA-3 score (.40% of weighted observations), and 44 missing a PSS-4 score (.72% of weighted observations).
[a]Scores for the Perceived Stress Scale (PSS-4) were dichotomized at the median.

vs. 54%), loneliness (67% vs. 61%), and greater perceived stress (48% vs. 31%). Black/African American, Hispanic, and other/multiple race students were more likely than White and Asian students to report clinically significant depressive symptoms (66% vs. 63%) and greater perceived stress (44% vs. 40%). Clinically significant depressive symptoms varied by age group, with students 21–24 years reporting the highest prevalence (67%) and lower estimates in the other age groups (<21 years = 62%, 25–34 years = 63%, and ≥35 years = 59%). Further, students <21 and 21–24 years were more likely than students 25–34 and ≥35 years to report loneliness (70% vs. 51%) and greater perceived stress (43% vs. 36%). Lastly, undergraduates were more likely than graduate and professional students to report loneliness (70% vs. 55%) and greater perceived stress (43% vs. 37%).

Self-isolation was associated with the prevalence of clinically significant depressive symptoms, loneliness, and greater perceived stress, such that a higher relative prevalence was observed for each increase in level of self-isolation (Fig 1). For example, compared to students self-isolating none of the time, students self-isolating some of the time were 1.78 times as likely to have clinically significant depressive symptoms (95% CI 1.37, 2.30). Further, students self-isolating most or all of the time were 2.12 (95% CI 1.64, 2.74) and 2.27 (95% CI 1.75, 2.94) times as likely to have clinically significant depressive symptoms, respectively. Trends (p < .001) were observed between level of self-isolation and clinically significant depressive symptoms (Z = -25.76), loneliness (Z = -7.36), and greater perceived stress (Z = -11.36). We found similar associations between agreement with additional social isolation statements and greater psychological distress outcome prevalence (Table 4).

Supplemental materials contain unweighted exposure and outcome distributions (Tables 2.1 & 2.2 in S1 File) and expanded tables containing weighted and unweighted estimates for associations between all social isolation variables and psychological distress outcomes that include the total number of participants with a given outcome within each exposure category (Tables 3.1 & 3.2 in S1 File). Lastly, we provided weighted and unweighted estimates for associations between self-isolation and psychological distress outcomes stratified by age, race and ethnicity, gender, and student type (Tables 4.1 & 4.2 in S1 File).

## Discussion

The prevalence of psychological distress outcomes among a cohort of undergraduate, graduate, and professional students in the southern U.S. in June 2020 was strikingly high. Clinically

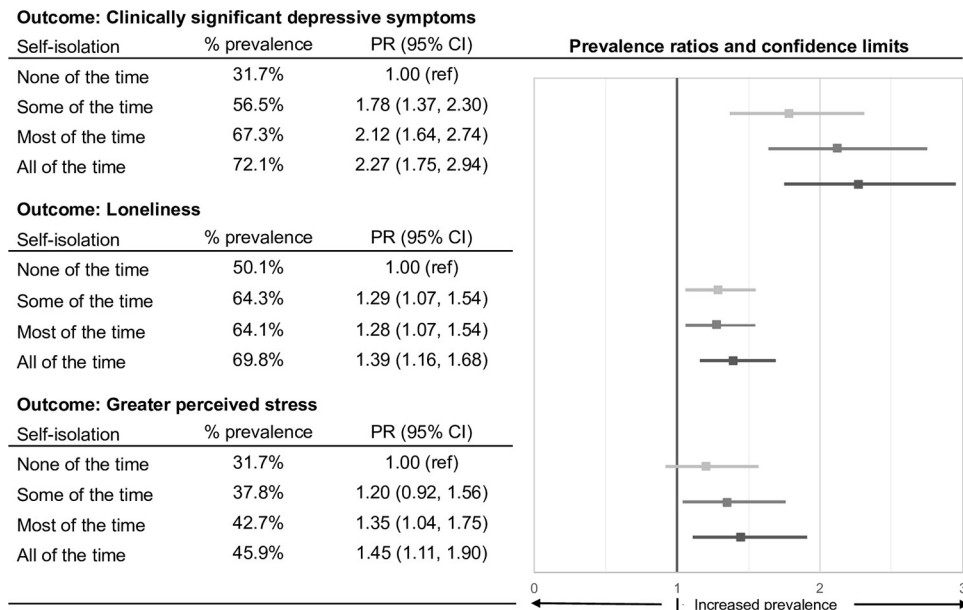

**Outcome: Clinically significant depressive symptoms**

| Self-isolation | % prevalence | PR (95% CI) |
|---|---|---|
| None of the time | 31.7% | 1.00 (ref) |
| Some of the time | 56.5% | 1.78 (1.37, 2.30) |
| Most of the time | 67.3% | 2.12 (1.64, 2.74) |
| All of the time | 72.1% | 2.27 (1.75, 2.94) |

**Outcome: Loneliness**

| Self-isolation | % prevalence | PR (95% CI) |
|---|---|---|
| None of the time | 50.1% | 1.00 (ref) |
| Some of the time | 64.3% | 1.29 (1.07, 1.54) |
| Most of the time | 64.1% | 1.28 (1.07, 1.54) |
| All of the time | 69.8% | 1.39 (1.16, 1.68) |

**Outcome: Greater perceived stress**

| Self-isolation | % prevalence | PR (95% CI) |
|---|---|---|
| None of the time | 31.7% | 1.00 (ref) |
| Some of the time | 37.8% | 1.20 (0.92, 1.56) |
| Most of the time | 42.7% | 1.35 (1.04, 1.75) |
| All of the time | 45.9% | 1.45 (1.11, 1.90) |

**Fig 1. Associations between level of self-isolation and psychological distress outcomes.** PR = prevalence ratio, CI = confidence interval; weighted PR estimates and 95% CIs were calculated using log-binomial regression with a robust error variance; self-isolation: none of the time is the referent; scores for the Perceived Stress Scale (PSS-4) were dichotomized at the median.

**Table 4. Associations between other social isolation variables and psychological distress outcomes.**

| Social isolation variables | Depression[c] | | Loneliness | | Greater stress[d] | |
|---|---|---|---|---|---|---|
| | % | PR (95% CI) | % | PR (95% CI) | % | PR (95% CI) |
| **I avoid crowded areas[a]** | | | | | | |
| Somewhat/strongly disagree | 46.7 | 1.00 (ref) | 59.2 | 1.00 (ref) | 38.0 | 1.00 (ref) |
| Somewhat/strongly agree | 64.6 | 1.38 (1.19, 1.61) | 64.9 | 1.10 (0.97, 1.23) | 41.3 | 1.09 (0.91, 1.30) |
| **I avoid getting together with people who are not part of my household[a]** | | | | | | |
| Somewhat/strongly disagree | 56.6 | 1.00 (ref) | 65.1 | 1.00 (ref) | 40.1 | 1.00 (ref) |
| Somewhat/strongly agree | 66.1 | 1.17 (1.11, 1.23) | 64.1 | 0.98 (0.94, 1.03) | 42.0 | 1.05 (0.97, 1.13) |
| **I stayed home[b]** | | | | | | |
| Somewhat/strongly disagree | 45.9 | 1.00 (ref) | 53.7 | 1.00 (ref) | 37.0 | 1.00 (ref) |
| Somewhat/strongly agree | 65.3 | 1.42 (1.28, 1.58) | 65.6 | 1.22 (1.12, 1.34) | 44.6 | 1.13 (1.00, 1.28) |
| **I did not attend social gatherings[b]** | | | | | | |
| Somewhat/strongly disagree | 50.8 | 1.00 (ref) | 60.2 | 1.00 (ref) | 36.3 | 1.00 (ref) |
| Somewhat/strongly agree | 65.4 | 1.29 (1.18, 1.40) | 65.2 | 1.08 (1.01, 1.16) | 42.0 | 1.16 (1.04, 1.29) |

% = prevalence, PR = prevalence ratio, CI = confidence interval; weighted PR estimates and 95% CIs were calculated using log-binomial regression with a robust error variance; disagree/strongly disagree is the referent; participants with a missing a CES-D-10, UCLA-3, or PSS-4 score were excluded from relevant models; participants with missing scores for social isolation variables or who responded "neither agree nor disagree" were also excluded.

[a]Please indicate how much you disagree or agree with the following statements.

[b]To what extent do you agree with each of the following statements about your behavior in the past month as a result of the new coronavirus?

[c]Depression = clinically significant symptoms of depression (CES-D-10).

[d]Greater stress = perceived stress score above the median (PSS-4).

significant levels of symptoms of depression were reported by almost two-thirds of the students. Among samples of U.S. college students interviewed at a similar time during the pandemic, 32–48% screened positive for major depressive disorder [13] or showed a moderate-to-severe level of depression [12]. Although measures and criteria differ, global data has shown significant increases in depressive symptoms among all age groups due to the pandemic [23, 24]. For example, a study among U.S. adults found that depressive symptoms reported early in April-June 2020 were almost four times higher than before the pandemic (24.3% vs. 6.5%, respectively) [2, 25]. Another study found that the prevalence of depressive symptoms in the U.S. increased more than 3-fold during the COVID-19 pandemic, from 8.5% before COVID-19 to 27.8% during COVID-19 [26]. While we don't have estimates collected prior to the pandemic for comparison, the prevalence of clinically significant depressive symptoms in our sample was notably higher than in other investigations of college student mental health during the COVID-19 pandemic [9–15].

Among UNC-CH students, almost two-thirds were categorized as lonely. In a cross-cohort analysis of data from U.K. adults, 39% were categorized as lonely using the same UCLA-3 cut point during the pandemic compared to 26% before the pandemic [27]. There is consistent evidence linking loneliness to poor physical and mental health outcomes, particularly among young people [28–30]. For example, a rapid review on the impact of social isolation and loneliness on the mental health of children, adolescents, and young adults found that loneliness for long durations was associated with depression, anxiety, and posttraumatic stress [31]. Reducing feelings of loneliness is also crucial to preventing suicide [32–34]. Further, while our measure of perceived stress was dichotomized at the sample median, in other samples of U.S. college students, 38–39% of students screened positive for generalized anxiety disorder [13] or showed a moderate-to-severe level of anxiety, and 71% of students indicated their stress levels had increased during the pandemic [12].

Consistent with other studies conducted during the pandemic, adverse psychological distress outcomes in the current study were particularly pronounced among female students, Black/African American, Hispanic, and other/multiple race students, younger students, and undergraduate students [12–14]. These disparities are especially concerning given that other research has shown that college students of color have the lowest rates of mental health service utilization [35]. While we didn't collect data on family income, previous research has also demonstrated a higher prevalence of depression and anxiety among low-income students during the pandemic [13]. The same study found that financial concerns were the leading barrier to obtaining mental health care [13]. Given that the psychological impacts of the COVID-19 pandemic are expected to persist [36, 37], identifying those at heightened risk for severe mental health outcomes is critical so that effective mitigation strategies can be developed for ongoing responses to the pandemic and future disruptive events.

Our study found that level of self-reported self-isolation was associated with clinically significant depressive symptoms, loneliness, and greater perceived stress, with the largest estimates observed for depression. These findings are consistent with previous research that has demonstrated the profound impact of social isolation on mental health [28, 29, 38]. In the context of COVID-19, previous research has demonstrated the link between degree of social isolation and psychological distress among older adults [39]. The current study is among the first to establish this link among college students, a group that experiences disparate mental health outcomes compared to the general population [3–5]. In the era of COVID-19, social isolation is a widely shared experience. A study of social support and mental health among college students found that students with lower-quality social support were more likely to experience mental health problems [40]. School closures, abrupt transitions to remote learning, and social distancing measures implemented during the COVID-19 pandemic likely disturbed critical

social support systems among college students, exacerbating a co-epidemic of mental health symptoms and COVID-19 [41, 42].

Interventions to support students experiencing psychological distress during the ongoing COVID-19 pandemic are critical. In addition to reporting the highest prevalence of depression and anxiety disorders compared to any other age group early in the pandemic, young adults in the U.S. also reported the highest prevalence of substance use to cope with pandemic-related stress (25%) and suicidal ideation (26%) [2]. Mental health disorders can also negatively impact a student's academic success [14, 43], in addition to their general health and well-being. There are several barriers that limit the effectiveness of student mental health programs, including stigma, a lack of disability identity, and insufficient resources [44]. Colleges and universities should promote evidenced-based initiatives aimed at reducing the psychological impact of COVID-19, including counseling and psychological services and personal strategies to improve one's mental health (e.g., connecting with others, engaging in hobbies or physical activity, practicing meditation) [23, 45, 46]. Further, there is a growing body of research on evidenced-based interventions that can improve student mental health. For example, peer support interventions for depression have been found to be effective among university students [47–49], particularly among those who show low engagement in traditional mental health services, such as minority students [3, 50]. Last, to lessen the potential impacts of social isolation on student mental health, universities should prioritize facilitating opportunities for students to safely connect with their peers.

There are several limitations of this research. First, while this study used weighting to make inferences about the UNC-CH student population, findings may not be generalizable to U.S. college students more broadly. Second, we utilized clinically validated screening instruments to assess symptoms of mental health disorders and psychological distress; diagnostic evaluations were not conducted. Further, this study was cross-sectional, and thus, we do not have data collected prior to COVID-19 for comparison. Next, the survey administration coincided with protests in support of the Black Lives Matter movement across the U.S. This may have impacted student responses and confounded our analysis, given that Black/African American students reported a slightly higher prevalence of clinically significant depressive symptoms (67%). Lastly, although weighting methods were used to adjust for incomplete responses and nonresponse, their effectiveness is limited if there are differences between survey completers and those who partially completed or did not respond to the survey on study variables not accounted for by weighting.

## Conclusions

The prevalence of adverse mental health outcomes among public university students in the southern U.S. was exceptionally high, with 64% of students reporting clinically significant depressive symptoms. Given that college and university students represent approximately 6% of the U.S. population, these findings document a significant burden of psychological distress. Further, we found that adverse mental health outcomes were significantly associated with social isolation. Universities should expand access to clinical treatment service options and promote strategies for social connectedness and personal wellness. Research examining the long-term impacts of social isolation on mental health among college students and how universities can prepare systems to mitigate mental health consequences as the pandemic evolves and during future disruptive events is needed.

## Supporting information

**S1 File.**
(DOCX)

## Author Contributions

**Conceptualization:** Danielle Giovenco, Audrey Pettifor.

**Data curation:** Bryant Hutson, Laurie Buchanan, Audrey Pettifor.

**Formal analysis:** Danielle Giovenco, Bonnie E. Shook-Sa.

**Investigation:** Audrey Pettifor.

**Methodology:** Danielle Giovenco, Audrey Pettifor.

**Writing – original draft:** Danielle Giovenco.

**Writing – review & editing:** Danielle Giovenco, Bonnie E. Shook-Sa, Bryant Hutson, Laurie Buchanan, Edwin B. Fisher, Audrey Pettifor.

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
