## [Decision Letter · Decision Letter 0]

24 Apr 2022

PONE-D-22-04563Social isolation and psychological distress among southern US college students in the era of COVID-19PLOS ONE

Dear Dr. Giovenco,

Thank you for submitting your manuscript to PLOS ONE. After careful consideration, we feel that it has merit but does not fully meet PLOS ONE’s publication criteria as it currently stands. Therefore, we invite you to submit a revised version of the manuscript that addresses the points raised during the review process.

We look forward to receiving your revised manuscript.

Kind regards,

Carlos Miguel Rios-González, Ph.D

Academic Editor

PLOS ONE

Journal Requirements:

(This research was supported by the University of North Carolina at Chapel Hill and grant F31MH119965 (PI: Giovenco) of the National Institutes of Mental Health. The funders had no role in study design, data collection and analysis, decision to publish, or preparation of the manuscript.)

Reviewers' comments:

Reviewer's Responses to Questions

**Comments to the Author**

1. Is the manuscript technically sound, and do the data support the conclusions?

Reviewer #1: Partly

2. Has the statistical analysis been performed appropriately and rigorously? 

Reviewer #1: No

3. Have the authors made all data underlying the findings in their manuscript fully available?

Reviewer #1: Yes

4. Is the manuscript presented in an intelligible fashion and written in standard English?

Reviewer #1: Yes

5. Review Comments to the Author

Reviewer #1: The manuscript addresses a timely topic and one appropriate to the journal. There are a number of strengths, including use of a sample that included a range of college student levels and provided a good bit of statistical power to work with. It is largely accessibly written (with some clarifications noted below recommended, and capable analyses was undertaken. The manuscript also has several areas where fuller elaboration and clarification would be useful. The following questions and suggestions are offered in a constructive spirit to help illuminate areas to particularly consider focus on.

• The introduction makes a range of legitimate, straightforward points. My concern is how to help readers understand early on how this paper provides complementary or value-added findings, especially coming (likely) in late 2022. There has been such a surge of reports (very much including college students) as to psychological struggles during the pandemic and the relationship of various aspects of isolation to mental health. I suggest a fuller reporting of the literature in this arena, and setting up with greater specificity how your findings extend on this.

• Sounds like there could be a good bit of missing data if students skipping 20-25% if item responses are included. Yet, as I read the table notes at bottom (eg, Table 3), it looks like missingness was low (eg, .63% of sample). How does this square with earlier presented missingness information? It will be best to provide a clearer summary in the measurement section as to what percentage of the sample did not have their own responses for each of the psychological variables used.

• Then these measure values went through the raking process. I understand the logic of using data from fully compliant respondents to create the measures. Yet this also raises some questions about the validity of those psych variable values for students who were not fully responsive. Did the authors consider a combination of data imputation for missingness alongside raking for sample weighting to better match the population characteristics?

• Did students whose psychological variable values came from fully compliant students vary on other key variables—eg, sociodemographics? Isolating? This raises questions for interpretation of findings. At a minimum, this must be discussed in the limitations.

• As I just now read a sentence that missing data were minimal with <2% missing doesn’t seem to add up to other missingness information presented. Please clarify this. just that just pertain to one item?

• Please provide a rationale for dichotomizing the psychological measures. Rather than using the full ranges of variance was the aim to establish equivalents of clinical cutting scores?

• The literature has pointed out the importance of student financial resources and where they sheltered during the online period (eg, doing college from home, especially for first gen or lower income students). Not surprisingly, lower income students suffered more greatly re isolation and mental health. Students who needed to move back home with families also had a more difficult time. Possible to include these student characteristics in analysis? I strongly encourage student or family income to be used or some proxy.

• I was a little surprised to be see analysis remain at a fairly descriptive and often bivariate level. Given how much we know about isolation during COVID and its erosive associations with psychological functioning, providing a more theorized approach is likely to be useful. Why, for example, do the younger college students seem to be reporting greater struggle? Some research is using social determinant, multiple minority statuses, and/or developmental theorizing to assess how all of this affects mental health within multivariate frameworks.

• This richer introductory set up may then support a richer theorized discussion. The discussion notes a range of related findings and of steps that colleges/universities have been taking to reduce isolation, to increase access to supports and reduce stigma/barriers, to foster resilience and positive coping. Sharpen your message as to how your findings provide interesting extension of such points. Do you think there are ongoing shifts that will be helpful to students and student supports as we are more fully re-engaging?

My best wishes on the authors’ continued work.

6. PLOS authors have the option to publish the peer review history of their article (what does this mean?). If published, this will include your full peer review and any attached files.

Reviewer #1: No

---

## [Author Response · Author response to Decision Letter 0]

16 Jun 2022

JOURNAL REQUIREMENTS:

Response: We have review the style requirements and ensured they have been incorporated into our revised manuscript. 

Response: We have added that “written electronic consent was obtained from all participants” and “students had to be at least 18 years of age to be eligible to participate.” 

(This research was supported by the University of North Carolina at Chapel Hill and grant F31MH119965 (PI: Giovenco) of the National Institutes of Mental Health. The funders had no role in study design, data collection and analysis, decision to publish, or preparation of the manuscript.)

Response: We have edited the funding statement: “This research was supported by the University of North Carolina at Chapel Hill and grant F31MH119965 (PI: Giovenco) of the National Institutes of Mental Health. The funders had no role in study design, data collection and analysis, decision to publish, or preparation of the manuscript. There was no additional external funding received for this study.” This has been added to the cover letter and title page. 

REVIEWER COMMENTS

Reviewer #1: The manuscript addresses a timely topic and one appropriate to the journal. There are a number of strengths, including use of a sample that included a range of college student levels and provided a good bit of statistical power to work with. It is largely accessibly written (with some clarifications noted below recommended, and capable analyses was undertaken. The manuscript also has several areas where fuller elaboration and clarification would be useful. The following questions and suggestions are offered in a constructive spirit to help illuminate areas to particularly consider focus on.

Response: Thank you for your thorough review and constructive comments.

Comment: The introduction makes a range of legitimate, straightforward points. My concern is how to help readers understand early on how this paper provides complementary or value-added findings, especially coming (likely) in late 2022. There has been such a surge of reports (very much including college students) as to psychological struggles during the pandemic and the relationship of various aspects of isolation to mental health. I suggest a fuller reporting of the literature in this arena, and setting up with greater specificity how your findings extend on this.

Response: We agree the introduction needed to be updated. We have updated the literature in this area and formatted the last paragraph to describe specifically what our findings add. 

Comment: Sounds like there could be a good bit of missing data if students skipping 20-25% if item responses are included. Yet, as I read the table notes at bottom (eg, Table 3), it looks like missingness was low (eg, .63% of sample). How does this square with earlier presented missingness information? It will be best to provide a clearer summary in the measurement section as to what percentage of the sample did not have their own responses for each of the psychological variables used.

Response: Only students who completed the survey (i.e., made it to the end of the survey even if items were missed) were included in the analysis. We have clarified this in the manuscript. Since the mental health questions were at the end of the survey, the majority of students who did not complete the survey were missing these outcomes. By removing those who partially completed the surveys, were had minimal missing data in our analytic sample. To examine the potential for bias due to excluding persons who started but did not complete the surveys, we compared the distribution of demographic characteristics, self-isolation, and psychological distress outcomes for survey completers (included in the analysis) and all survey respondents and found that the groups were similar on all measures examined. This information has been added to the supplement (S1 Table 1.1) and is also described in the statistical analysis and results sections. 

Comment: Then these measure values went through the raking process. I understand the logic of using data from fully compliant respondents to create the measures. Yet this also raises some questions about the validity of those psych variable values for students who were not fully responsive. Did the authors consider a combination of data imputation for missingness alongside raking for sample weighting to better match the population characteristics?

Response: In our analysis, students with partially completed surveys are treated similarly as students who did not respond to the survey. The raking accounts for differences in demographic characteristics between those who completed the survey and those who partially completed or did not respond to the survey. In the limitations, we explain that, “although weighting methods were used to adjust for incomplete responses and nonresponse, their effectiveness is limited if there are differences between survey completers and those who partially completed or did not respond to the survey on study variables not accounted for by the weighting.”

Comment: Did students whose psychological variable values came from fully compliant students vary on other key variables—eg, sociodemographics? Isolating? This raises questions for interpretation of findings. At a minimum, this must be discussed in the limitations.

Response: We have added the following statement to the Results section: “A comparison of survey completers and all survey respondents on demographic characteristics and primary exposure and outcome variables revealed no substantive differences between groups (Supplement 1 Table 1.1).”

Comment: As I just now read a sentence that missing data were minimal with <2% missing doesn’t seem to add up to other missingness information presented. Please clarify this. just that just pertain to one item?

Response: We have edited this statement to be more specific: “Missing data for psychological distress variables in the analytic sample were minimal, with <2% of students missing data for measure items.”

Comment: Please provide a rationale for dichotomizing the psychological measures. Rather than using the full ranges of variance was the aim to establish equivalents of clinical cutting scores?

Response: We have added the statement: “When available, we dichotomized outcomes using clinically significant cutpoints to improve the interpretability of our findings.”

Comment: The literature has pointed out the importance of student financial resources and where they sheltered during the online period (eg, doing college from home, especially for first gen or lower income students). Not surprisingly, lower income students suffered more greatly re isolation and mental health. Students who needed to move back home with families also had a more difficult time. Possible to include these student characteristics in analysis? I strongly encourage student or family income to be used or some proxy.

Response: We unfortunately did not collect information on student or family income. We had added discussion of income from previous research to our Discussion section.

Comment: I was a little surprised to be see analysis remain at a fairly descriptive and often bivariate level. Given how much we know about isolation during COVID and its erosive associations with psychological functioning, providing a more theorized approach is likely to be useful. Why, for example, do the younger college students seem to be reporting greater struggle? Some research is using social determinant, multiple minority statuses, and/or developmental theorizing to assess how all of this affects mental health within multivariate frameworks.

Response: We have made substantial revisions to the Discussion to provide more interpretation of our findings and a broader description of the existing literature. 

Comment: This richer introductory set up may then support a richer theorized discussion. The discussion notes a range of related findings and of steps that colleges/universities have been taking to reduce isolation, to increase access to supports and reduce stigma/barriers, to foster resilience and positive coping. Sharpen your message as to how your findings provide interesting extension of such points. Do you think there are ongoing shifts that will be helpful to students and student supports as we are more fully re-engaging?

Response: We have revised the Discussion section based on this feedback. Thank you.

---

## [Decision Letter · Decision Letter 1]

9 Nov 2022

PONE-D-22-04563R1Social isolation and psychological distress among southern US college students in the era of COVID-19PLOS ONE

Dear Dr. Giovenco,

Thank you for submitting your manuscript to PLOS ONE. After careful consideration, we feel that it has merit but does not fully meet PLOS ONE’s publication criteria as it currently stands. Therefore, we invite you to submit a revised version of the manuscript that addresses the points raised during the review process.

We look forward to receiving your revised manuscript.

Kind regards,

Md. Tanvir Hossain

Academic Editor

PLOS ONE

Reviewers' comments:

Reviewer's Responses to Questions

**Comments to the Author**

1. If the authors have adequately addressed your comments raised in a previous round of review and you feel that this manuscript is now acceptable for publication, you may indicate that here to bypass the “Comments to the Author” section, enter your conflict of interest statement in the “Confidential to Editor” section, and submit your "Accept" recommendation.

Reviewer #2: (No Response)

Reviewer #3: All comments have been addressed

Reviewer #4: All comments have been addressed

2. Is the manuscript technically sound, and do the data support the conclusions?

Reviewer #2: No

Reviewer #3: Yes

Reviewer #4: Yes

3. Has the statistical analysis been performed appropriately and rigorously? 

Reviewer #2: No

Reviewer #3: Yes

Reviewer #4: Yes

4. Have the authors made all data underlying the findings in their manuscript fully available?

Reviewer #2: Yes

Reviewer #3: Yes

Reviewer #4: Yes

5. Is the manuscript presented in an intelligible fashion and written in standard English?

Reviewer #2: (No Response)

Reviewer #3: Yes

Reviewer #4: Yes

6. Review Comments to the Author

Reviewer #2: (No Response)

Reviewer #3: The materials and method of empirical investigation was properly designed and performed. I would like to request the authors to incorporate the recommendations into the manuscript based on the findings because its capacity to disseminate ' what needs to be done' will scale up with greater heights.

Reviewer #4: This study aims to investigate the social isolation and psychological distress among southern US college students in the era of COVID-19.

Evidence suggests that the pandemic and the social isolation had a negative impact on mental health of the general public (particularly in college student ).

Limitations of the study are noted and discussed.

I think the paper covers an import area of research.

I have listed some specific comments below that the authors should take into account before this work could

Introduction:

The introduction should start with the global mental health effects of the COVID-19 lockdown among college students. Please add 2 or 3 references from different countries regarding the social isolation situation during COVID-19 among college students.

Page 03, line 51 and 52

“In addition to substance use to cope with pandemic related stress (25%) and suicidal ideation (26%), compared to any other age group”

This sentence doesn’t belong in the introduction. It should be in the result part.

1. Authors must be added references that work on social isolation among college students during Covid-19 in USA.

It would be useful for the authors to state more about mental health within USA so that the reader has a better understanding of local context. E.g., wide spread awareness of different mental health conditions, how youth would access support, ratio of mental health professionals to population. This will also help link to the discussion later also.

Method:

Method section is well described.

Just clarify one thing. In line 164- 1. what do you mean by clinically significant depressive symptoms. Could you please mention 2 or 3 symptoms?

2.please add a flow chart to represent your sampling technique.

3. Was the study pre-registered anywhere?

Result:

1. Please clarify the sample is the mean age of the participants, the previous mental state and the taking of medication.

2. Could you please clarify the word self-isolation

Discussion:

Though a summary of the main findings is provided, I felt that overall, there could be more consideration of implications. For example, what is the current policy on mental health in young people and available support and what do the study findings mean about the adequacy of this provision/infrastructure.

7. PLOS authors have the option to publish the peer review history of their article (what does this mean?). If published, this will include your full peer review and any attached files.

Reviewer #2: No

Reviewer #3: **Yes: **Abul Hasan BakiBillah

Reviewer #4: **Yes: **Sadia Afrin

---

## [Author Response · Author response to Decision Letter 1]

11 Nov 2022

November 11, 2022

RE: PLOS ONE RESUBMISSION

Dear Editorial Team,

We appreciate the opportunity to provide additional revisions on our manuscript. We have addressed all the reviewers’ comments below and in the revised manuscript. 

Reviewers' comments to the author:

Reviewer #2: (No Response)

Reviewer #3: The materials and method of empirical investigation was properly designed and performed. I would like to request the authors to incorporate the recommendations into the manuscript based on the findings because its capacity to disseminate 'what needs to be done' will scale up with greater heights.

Reviewer #4: This study aims to investigate the social isolation and psychological distress among southern US college students in the era of COVID-19.

Evidence suggests that the pandemic and the social isolation had a negative impact on mental health of the general public (particularly in college student ).

Introduction:

Comment: The introduction should start with the global mental health effects of the COVID-19 lockdown among college students. Please add 2 or 3 references from different countries regarding the social isolation situation during COVID-19 among college students.

Response: Thank you for the suggestion. We have added several references to the line “There is a growing body of literature describing college student mental health in the context of the COVID-19 pandemic” (paragraph 3, lines 63-64), including 3 citations from global studies and 4 domestic citations. We have decided to keep the beginning of the Introduction as a description of mental health outcomes resulting from the pandemic among young adults more broadly. 

Comment: Page 03 lines 51 and 52 “In addition to substance use to cope with pandemic related stress (25%) and suicidal ideation (26%), compared to any other age group”. This sentence doesn’t belong in the introduction. It should be in the result part.

Response: This line presents finding from the cited study, not findings from our own work. We have, however, moved this to the Discussion. 

Comment: Authors must be added references that work on social isolation among college students during Covid-19 in USA. It would be useful for the authors to state more about mental health within USA so that the reader has a better understanding of local context. E.g., wide spread awareness of different mental health conditions, how youth would access support, ratio of mental health professionals to population. This will also help link to the discussion later also.

Response: We have expanded the Introduction, which now includes several additional references related to mental health among students in the US to provide more context. 

Methods:

Comment: Method section is well described.

Comment: Just clarify one thing. In line 164- 1. what do you mean by clinically significant depressive symptoms. Could you please mention 2 or 3 symptoms?

Response: This is described in “psychological distress outcomes” sections. We utilize the CES-D-10, which includes three items on depressed affect, five items on somatic symptoms, and two on positive affect. We also provide a citation for a previous study that has utilized the cut point of 10 for clinically significant symptoms of depression and where measure items can be found. 

Comment: Please add a flow chart to represent your sampling technique.

Response: Students were not sampled for this study, but, instead, all UNC-CH undergraduate, graduate, and professional students received the survey. Then, to adjust for student nonresponse (or partial response), we used iterative proportional fitting (i.e., raking) methods to weight the sample of survey completers to the marginal distributions of the UNC-CH student population by age category, race and ethnicity, gender, and student type. 

Comment: Was the study pre-registered anywhere?

Response: It was not. 

Results:

Comment: Please clarify the sample is the mean age of the participants, the previous mental state and the taking of medication.

Response: We have added the median age of participants in the unweighted sample to the Results section. We do not have data on previous mental state or medication use. 

Comment: Could you please clarify the word self-isolation

Response: We used the term social isolation to describe the combination of exposure measures we examined, including a self-isolation question and four additional questions where students were asked the extent to which they agree with statements about avoiding social behaviors. This information can be found in the Measures sub-section of the Methods.

Discussion:

Comment: Though a summary of the main findings is provided, I felt that overall, there could be more consideration of implications. For example, what is the current policy on mental health in young people and available support and what do the study findings mean about the adequacy of this provision/infrastructure.

Response: Mental health policy and available support varies widely between institutions in the US. However, we have added more information to the Discussion about the barriers that limit the effectiveness of student mental health services, including stigma, a lack of disability identity, and insufficient resources. We also have elaborated on how these programs should respond to the pandemic and incorporate evidence-based interventions to improve student mental health.

---

## [Decision Letter · Decision Letter 2]

8 Dec 2022

Social isolation and psychological distress among southern U.S. college students in the era of COVID-19

PONE-D-22-04563R2

Dear Dr. Giovenco,

We’re pleased to inform you that your manuscript has been judged scientifically suitable for publication and will be formally accepted for publication once it meets all outstanding technical requirements.

Kind regards,

Md. Tanvir Hossain

Academic Editor

PLOS ONE

Reviewers' comments:

Reviewer's Responses to Questions

**Comments to the Author**

1. If the authors have adequately addressed your comments raised in a previous round of review and you feel that this manuscript is now acceptable for publication, you may indicate that here to bypass the “Comments to the Author” section, enter your conflict of interest statement in the “Confidential to Editor” section, and submit your "Accept" recommendation.

Reviewer #3: All comments have been addressed

Reviewer #4: All comments have been addressed

2. Is the manuscript technically sound, and do the data support the conclusions?

Reviewer #3: Yes

Reviewer #4: Yes

3. Has the statistical analysis been performed appropriately and rigorously? 

Reviewer #3: Yes

Reviewer #4: Yes

4. Have the authors made all data underlying the findings in their manuscript fully available?

Reviewer #3: Yes

Reviewer #4: Yes

5. Is the manuscript presented in an intelligible fashion and written in standard English?

Reviewer #3: Yes

Reviewer #4: Yes

6. Review Comments to the Author

Reviewer #3: After incorporating the comments into the revised manuscript, Now, it is more scientifically sound and bears a powerful message to the policy making table for propelling public health interventions in saving innocent younger generation.

Reviewer #4: Satisfied with author's response. They were addressed all the comments with proper answer and references.

7. PLOS authors have the option to publish the peer review history of their article (what does this mean?). If published, this will include your full peer review and any attached files.

Reviewer #3: **Yes: **Abul Hasan BakiBillah

Reviewer #4: **Yes: **Sadia Afrin

---

## [Editor Report · Acceptance letter]

22 Dec 2022

PONE-D-22-04563R2 

Social isolation and psychological distress among southern U.S. college students in the era of COVID-19 

Dear Dr. Giovenco:

I'm pleased to inform you that your manuscript has been deemed suitable for publication in PLOS ONE. Congratulations! Your manuscript is now with our production department. 

Kind regards, 

on behalf of

Dr. Md. Tanvir Hossain 

Academic Editor

PLOS ONE